# Forgive, Let Go, and Stay Well! The Relationship between Forgiveness and Physical and Mental Health in Women and Men: The Mediating Role of Self-Consciousness

**DOI:** 10.3390/ijerph20136229

**Published:** 2023-06-26

**Authors:** Justyna Mróz, Kinga Kaleta

**Affiliations:** Department of Psychology, Jan Kochanowski University of Kielce, 25-029 Kielce, Poland; kinga.kaleta@ujk.edu.pl

**Keywords:** forgiveness, self-consciousness, rumination, mental health, physical health, gender differences

## Abstract

Background: The current study assessed forgiveness (positive forgiveness and reduced unforgiveness), self-consciousness (rumination and reflection), and physical and mental health. The aim of the study was to check if self-consciousness mediates the relationship between dispositional forgiveness and health. Methods: To address this link, we conducted 2 studies (*N* = 484 in Study 1 and *N* = 249 in Study 2). Data were collected separately for Study 1, as well as Study 2. We used in both studies the Heartland Forgiveness Scale and the Rumination-Reflection Questionnaire, and additionally, the General Health Questionnaire-28 in Study 1 and the Scales of Psychological Well-Being in Study 2. Results: The results indicated that rumination was an effective mediator between positive forgiveness and mental health (B = 0.14, CI_95%_ = [0.064, 0.234]), reduced unforgiveness and physical health (B = −0.13, CI_95%_ = [−0.182, −0.088]), and both rumination and reflection between reduced unforgiveness and mental health (B = 0.13 CI_95%_ = [0.051, 0.226]), positive forgiveness (B = −0.09 CI_95%_ = [−0.135, −0.052]), and physical health. Conclusion: The mediating role of ruminations was more frequently observed in females. The study highlighted the indirect effect between forgiveness and health. Rumination and reflection are mediators between forgiveness and health.

## 1. Introduction

Psychological predictors of mental health are constantly sought after. One of them is forgiveness, suggested as an important component of health. The relationships between forgiveness and health have gained increasing empirical attention since forgiveness may have many salutary benefits [1,2,3,4]. Most definitions of forgiveness focus on the transformation of negative emotions, thoughts, or behaviors toward the offender into neutral or even positive ones. Enright defined forgiveness as “a willingness to abandon one’s right to resentment, negative judgment, and indifferent behavior toward one who unjustly injured us, while fostering the undeserved qualities of compassion, generosity and even love toward him or her” [5] (pp. 46–470). Forgiveness also helps one cope with negative outcomes of damaging situations, which contributes to better well-being [6,7]. Some scholars highlight both the positive and the negative dimension of forgiveness [8,9,10]. The positive dimension of forgiveness, called positive forgiveness, focuses on the benevolent and conciliatory motivation toward the wrongdoer, or the desire to be generous and kind toward them [10]. The negative dimension of forgiveness involves transforming beliefs, emotions, and motivations towards the offender from negative to neutral. These changes are manifested by a decreased willingness to take revenge, to avoid, and to feel anger, resentment, passion, reluctance, hate, or bitterness and, therefore, are called reduced unforgiveness. Positive forgiveness and reduced unforgiveness have different predictors [9,11]. Positive forgiveness is related to agreeableness, whereas reduced unforgiveness is inversely related to neuroticism [9]. Kaleta and Mróz [12] found positive correlations between positive forgiveness and satisfaction with life, but primarily among older people, and between reduced unforgiveness and satisfaction with life among younger adults.

Although numerous studies confirm the relation between forgiveness and health [2,3,4], few have examined the differences between positive forgiveness, reduced unforgiveness, and dimensions of health. Thus, the psychological mechanism linking positive forgiveness, reduced unforgiveness, and health is not clear enough. Forgiveness helps to cope with the experience of harm [7] and can be enhanced by, among other things, self-consciousness. According to previous research, there is a correlation between forgiveness and a decrease in ruminative thoughts [13]. Thus, forgiveness may reduce repetitive thoughts and enhance reflection, leading to better health. The purpose of this research was to test how positive forgiveness and reduced unforgiveness relate to health. Additionally, we explored self-consciousness as a potential mediator accompanying this relationship.

### 1.1. Forgiveness and Physical and Mental Health

The relationship between forgiveness and health is reasonably well documented, both for physical [3] and mental health [4]. As regards the relationship between forgiveness and mental health, it was found that the presence of positive forgiveness is positively associated with existential well-being [14], life satisfaction [6], and life satisfaction among older adults [12]. On the other hand, reduced unforgiveness is negatively associated with unhealthy outcomes (depression, anxiety, somatic symptoms, social dysfunction) and perceived stress [6], depression, and anger [14]. Further, Chung [15] showed there is a positive correlation between unforgiveness and depression. Similar results were obtained in other studies. For example, research showed that older people who are more forgiving are also less depressive and with lower levels of stress than individuals who find it difficult to forgive [4]. In Christian women who had experienced abuse, higher levels of forgiveness were found to be related to better mental health and inversely related to depression [16].

To sum up, in the relationship between forgiveness and health, the propensity to forgive is associated with perceiving one’s well-being as high. Considering the division between positive forgiveness and reduced unforgiveness, we expect, based on earlier studies, the existence of a positive correlation between positive forgiveness (more benevolence, warmth, compassion, etc. toward the offender) and mental health measures that bring positive outcomes (e.g., psychological well-being). On the other hand, we also expect reduced unforgiveness (lower tendency to seek revenge and lower levels of anger, hate, etc. toward the offender) to have more significant, inverse relationships with negative health aspects, such as depression or anxiety. Thus, the tendency to overcome one’s negative attitude toward the offender contributes to lowering the severity of symptoms related to poor health. Previous studies confirm the relationship between forgiveness and health [4,15,16]; however, the mechanism clarifying this link is still investigated. For example, Zhu [17] found that forgiveness is linked with greater satisfaction with life via social support. Further, this path was verified among older people by Tian and Wang [18]. Older people with a stronger tendency to forgive were likely to be involved in more social support, and consequently, their mental health/well-being was greater. In another study, cynicism and psychache mediated the relationship between forgiveness and suicidal behavior [19]. Forgiveness was associated with less suicidal behavior through less cynicism and less psychache. Finally, Webb et al. [20] found that both forgiveness and mindfulness support better health (physical health, somatic symptoms, mental health, and psychological distress). Previous studies indicate that many variables can mediate the link between forgiveness and health. To the best of our knowledge, the indirect role of self-consciousness has not been studied so far.

### 1.2. Self-Consciousness as a Mediator

Self-consciousness is linked to positive, as well as negative, experiences and personality traits; Trapnell and Campbell call it the self-absorption paradox [21]. According to the researchers, self-consciousness has two subtypes that distinguish the maladaptive from the adaptive aspects of self-focus: (1) rumination, which represents the tendency to dwell on, rehash, and reevaluate events or experiences, and (2) reflection, defined as “intellectual self-attentiveness” through exploration, analysis, and contemplation of the self [21]. Additionally, Thomsen et al. [22] (p. 106) see “rumination as an unconstructive form and reflection as a constructive form of self-focused repetitive thoughts”. Rumination is motivated by fear and involves involuntary and repetitious concentration on one’s own experience, essentially having negative undertones. Reflection, on the other hand, is motivated by curiosity to get to know oneself [21]. The Goal Progress Theory describes rumination as the tendency to think repetitively about incomplete experiences, regardless of the valency of the thoughts [23]. Thus, rumination is an example of the Zeigarnik effect, when people think repeatedly about the wrongdoing that is unforgiven or unpunished. In the concept of self-consciousness proposed by Trapnell and Campbell, the thoughts about an incomplete task can be negative (rumination) and more positive and neutral (reflection) [21].

Gender differences in addressing the sources of distress through self-consciousness have been examined previously. Females are more likely to ruminate than males [24,25], particularly on a stress-related problem because of their role, for example [26]. According to the response styles theory [24], females tended to find emotional experiences and social events more intense and uncontrollable than males, and females were more likely to feel responsible for difficult events they experience, which might escalate ruminative thoughts [25,27].

A considerable body of research shows relationships between rumination, forgiveness, and health [28,29]. Some empirical studies consistently show that forgiveness is inversely correlated with rumination [28,30]; thus, people with a higher tendency to forgive have less intrusive, repetitive thoughts. With regard to the link between rumination and health, a maladaptive role of rumination was found. Rumination was found to be related to perceived chronic stress and unhealthy outcomes, such as depression, poor sleep/sleeping difficulty, anxiety [29], and increased depression [31]. Stoia-Caraballo et al. [32] showed that forgiveness is linked to a less negative affect and less anger rumination, which in turn is related to a better night’s sleep. On the other hand, Ingersoll-Dayton et al. [33] found that in older people, depressive symptoms correlate with rumination and unforgiveness.

Previous studies provide inconsistent insights into the relationship between reflection and health. While one study reported a negative association between reflection and depression [31], two separate studies indicated positive links between reflection (measured using the Ruminative Response Scale) and depression [34,35]. Additionally, Mori and Tanno [36] failed to identify this link. Thus, the precise nature of the association between reflection and health is not well understood.

In many studies, self-consciousness (rumination and reflection) was an important mediator [37,38,39]. For example, rumination (self-reproach, contemplation, depressive rumination) mediated the effect of childhood trauma on depression and anxiety in a non-clinical population [39]. Further, Jarukasemthawee and Pisitsungkagarn [38] showed the mediating role of rumination for mindfulness and eudemonic well-being: mindfulness reduced rumination and enhanced well-being.

Because previous studies show self-consciousness (rumination and reflection) to be related to both forgiveness and health [28,30,40], these variables can theoretically also be considered as mediators between forgiveness and health. Martin et al. [23] analyzed self-focus on repetitive thoughts referring to the Goal Progress Theory of rumination and the Zeigarnik effect, showing that uncompleted tasks in the action loop are remembered, whereas forgiveness can help close the loop. It is similar to a cognitive rehearsal when the victim ruminates on the offender and the hurtful incident, and it is part of the forgiveness process proposed by Enright [5]. Thus, forgiveness may inhibit rumination and enhance reflection, for example, [13] which may lead to better mental and physical health, for example [39]. This study will investigate if self-consciousness mediates between forgiveness and health. With this, we can fill the research gap in this area.

### 1.3. Aim of the Study

This study aimed to examine the relationship between forgiveness and health. In previous studies, scholars found a link between positive forgiveness, reduced unforgiveness, and mental health (depression, anxiety, somatic symptoms, social dysfunction, psychological well-being) [12,41].

In this study, we also focused on the role of rumination and reflection as protective or inhibitive factors in the relationship between forgiveness and health. Previous studies indicated the mediating role of rumination and reflection on health and well-being [37,38,39]. To our knowledge, self-consciousness as a mediator of the relationship between forgiveness and health has never been analyzed.

We verified the hypotheses in two studies, so that we could see whether self-consciousness would be a universal mediator, regardless of the type of health presented by the dependent variable. We put forward the following main hypotheses:

**Hypothesis** **1.**
*A considerable body of literature [4,9] suggests there is a relationship between forgiveness and health, which means that positive forgiveness and reduced unforgiveness will be inversely related to the intensity of unhealthy outcomes (somatic problems, anxiety, social dysfunction, and depression) and positively related to mental health (intensity of psychological well-being).*


**Hypothesis** **2.**
*Referring to previous studies [39,42] and theories (e.g., of the Goal Progress Theory of rumination), we expect rumination and reflection will mediate the relationship between forgiveness and health (intensity of unhealthy outcomes and psychological well-being), with rumination acting as an inhibitor and reflection as a protector in this relationship. Considering the response styles theory [24] and previous studies [25,26,27], we conclude that there will be a difference in the mediating role of rumination and reflection among females and males (H2A).*


The hypotheses were tested in two studies.

## 2. Materials and Methods

### 2.1. Participants

#### 2.1.1. Study 1

Data were collected from 485 adults (74% females) aged 18 to 79 (*M* = 29.3; *SD* = 11.68), all of whom were Polish. They usually lived in cities (46.4%), or in the country (37.7%), and less often in towns (15.7%). As regards the level of education, 50.7% of participants completed secondary education, 42.7% higher education, and 6.4% a college education. In terms of marital status, 65.4% were single, 30.5% were married, 4% were widowed, and 3.5% were divorced. We asked students from our university to take part in the study. Moreover, each student was asked to recruit adults from different age groups. The respondents voluntarily agreed (without remuneration) to participate in the study. They had to complete three questionnaires: the Heartland Forgiveness Scale, the Rumination-Reflection Questionnaire, and the General Health Questionnaire-28.

#### 2.1.2. Study 2

The sample consisted of 249 individuals. The mean age of the participants was M = 33.36 (*SD* = 12.38), ranging from 18 to 77 years of age; 70% of the sample were women. The majority of respondents lived in the city (49.8%), 27.3% were town residents, and 22.9% lived in the country. The educational level ranged from vocational education (9.3%), through secondary (44.4%), and college (8.8%), to higher education (35.8%). Finally, 42% were married, 46.3% were single, 3.4.% were widowed, and 6.3% were divorced. All participants met two inclusion criteria: they were adults and had serious adverse childhood experiences (ACSs). The ACS was assessed via 11 questions with yes-no response categories designed for traumatic events during childhood. The participants were asked about neglect, physical/emotional abuse, and parental alcohol use. They were invited to continue the study if they answered yes to at least one question. They were recruited by trained psychology students. The respondents completed three questionnaires: the Heartland Forgiveness Scale, the Rumination-Reflection Questionnaire, and the Scales of Psychological Well-being.

### 2.2. Measures

In Study 1 and Study 2, we measured forgiveness and self-consciousness.

Disposition to forgive was measured using the Polish adaptation [43] of the Heartland Forgiveness Scale [44]. HFS is a multidimensional tool assessing dispositional forgiveness of oneself, others, and situations beyond anyone’s control. Participants rate their responses to 18 items using a 7-point scale (ranging from absolutely false to absolutely true). Sample items include: “With time I am understanding of myself for mistakes I’ve made”; “If others mistreat me, I continue to think badly of them”; “I eventually make peace with bad situations in my life”. The original version consists of three subscales (Forgiveness of Self, Forgiveness of Others, and Forgiveness of Situations). The Polish version comprises two scales measuring forgiveness in two separate domains—(1) the negative N scale, measuring the reduction of hostile thoughts, feelings, and behaviors (namely, overcoming unforgiveness) and (2) the positive P scale, measuring benevolent thoughts, feelings, and behaviors—and six subscales with the distinction between forgiveness of self, others, and situations (N—self, N—others, N—situations, P—self, P—others, P—situations). Higher scores on a particular subscale reflect a higher level of forgiveness in a particular area. The total HFS score indicates how forgiving a person tends to be. In the present study, we used the total HFS and Positive and Negative Scales only, and the values of Cronbach’s alpha ranged from 0.75 to 0.81 in Study 1 and from 0.81 to 0.85 in Study 2.

Self-consciousness. The Rumination-Reflection Questionnaire (RRQ) by Trapnell & Campbell [21] measures two different forms of self-consciousness: rumination and reflection. The former measures individuals’ tendency to repeatedly self-focus on past actions, whereas the latter measures the philosophical love of self-exploration [21]. The RRQ contains self-rumination and self-reflection subscales. We used the Polish version of Carter’s modified version of the RRQ [45]. The tool includes 13 items (e.g., “My attention is often so focused on the different characteristics of my person that I cannot get away from it.”) rated on a 5-point scale from 1 (*I do not agree*) up to 5 (*I agree completely*). The questionnaire obtained satisfactory psychometric parameters, although they were lower than in the original version. In study 1, Cronbach’s α was 0.77 for rumination and 0.65 for reflection, whereas in study 2, 0.67 and 0.60, respectively.

In addition, in Study 1 we measured problems with health. The General Health Questionnaire-28 (GHQ-28) by Makowska et al. [46] is a tool widely used to estimate the likelihood of participants having a health disorder. The questions are scored using the Likert method of 1-2-3-4. The questionnaire yields four subscales: somatic symptoms, anxiety and insomnia, social dysfunction, and severe depression. Cronbach’s alpha ranged from 0.80 to 0.86.

In Study 2, we measured mental health as psychological well-being. Psychological well-being was measured using the shortened version of the Scales of Psychological Well-being [47] (Polish version [48]), which includes 18 items (e.g., “For me, life has been a continuous process of learning, changing, and growth”) rated on a scale from 1 (*strongly disagree*) to 6 (*strongly agree*). The tool comprises six subscales: Autonomy, Environmental Mastery, Personal Growth, Positive Relations with Others, Purpose in Life, and Self-Acceptance, with three items per subscale. Participants responded to items using a 7-point Likert scale from 1 (strongly disagree) to 7 (strongly agree). For the present studies, Cronbach’s alpha ranged from 0.60 to 0.83.

### 2.3. Data Analyses

The preliminary analyses included calculating Pearson’s zero-order correlations, performed with IBM SPSS Statistics 26 (PS IMAGO PRO 6.0, Predictive Solutions, Krakow, Poland). Next, we tested the mediation model with multiple mediators (model 4) using PROCESS macro [49] through bootstrapping of 5000 subsamples at a confidence interval of 95%. The tendency to forgive was the predictor variable; rumination and reflection were entered as competing mediators; and general health (study 1) and psychological well-being (study 2) were the outcome variables. When the 95% confidence intervals for an indirect effect did not include zero, the indirect effect was significant [50]. We included age and gender as covariates, and we found both to be insignificant in both Study 1 and Study 2.

## 3. Results

### 3.1. Study 1

To verify the relationship among the tendency to forgive, rumination, reflection, and general health and its factors, we checked for any intercorrelations between the analyzed variables. The results are presented in Table 1.

Forgiveness (total score) and both positive forgiveness and reduced unforgiveness demonstrated inverse relationships with general scores, four subscales of unhealthy outcomes (somatic problems, anxiety, social dysfunction, and depression), and rumination. Additionally, positive forgiveness was positively related to reflection. Rumination showed a positive relationship with general scores and four subscales of unhealthy outcomes. Reflection was not related to health.

We tested multiple mediators simultaneously, with rumination and reflection as mediators between forgiveness (positive and reduced unforgiveness) and general health, with age and gender as covariates (see Figure 1 and Figure 2). Covariates were insignificant. We also tested this mediation with the general health subscales as dependent variables. All results shown are standardized. Positive forgiveness was associated with both mediators. Multiple mediation analyses showed that the total indirect effect of positive forgiveness on general health via rumination and reflection was significant (β = −0.09 CI_95%_ = [−0.135, −0.052]). Positive forgiveness was negatively associated with rumination (β = −0.17, *p* < 0.01) and positively associated with reflection (β = 0.17, *p* < 0.01). Rumination positively (β = 0.43, *p* < 0.001) and reflection negatively (β = −0.09, *p* < 0.05) predicted GHQ. Positive forgiveness had a significant negative total effect on general health (β = −0.21; *p* < 0.01). With the addition of rumination and reflection to the model, the direct effect of positive forgiveness on general health was also significant (β = −0.11, *p* < 0.05). The specific indirect effect of positive forgiveness on general health via rumination was significant (β =−0.07 CI_95%_ = [−0.116, −0.038]), and reflection was also significant (β = −0.02, CI_95%_ = [−0.057, −0.007]) (see Figure 1).

A similar effect was reported for anxiety/insomnia, where rumination was negatively and reflection positively predicted. The total effect was significant (β = −0.22, *p* < 0.05). After introducing rumination and reflection to the model, the total effect was not reduced to insignificant (DE β = −0.12, *p* = 0.05), indicating partial mediation. Positive forgiveness had an indirect effect on anxiety/insomnia through rumination (β = −0.06, CI_95%_ = [−0.105, −0.034]); reflection also was significant (β = −0.01, CI_95%_ = [−0.035, −0.001]). With regard to other dependent variables, positive forgiveness had a significant total effect on somatic problems (TE β = −0.02, *p* < 0.05). After introducing rumination and reflection to the model, the direct effect of positive forgiveness on somatic problems was not significant (DE β = −0.06), indicating full mediation. The specific indirect effect of positive disposition to forgive on somatic symptoms was significant through rumination (β = −0.047, CI_95%_ = [−0.079, −0.021]), but insignificant via reflection. Likewise, positive forgivingness had a significant total effect on social dysfunction (β = −0.12, *p* < 0.05), and after adding rumination and reflection to the model, the direct effect was not significant (β = −0.04), indicating full mediation via rumination and reflection. The specific indirect effect of a positive propensity to forgive on social dysfunction was significant through rumination (β = −0.052, CI_95%_ = [−0.086, −0.025]), but insignificant via reflection. On the other hand, positive forgiveness had a significant total effect on depression (β = −0.22, *p* < 0.001), and the direct effect was also significant (β = −0.14, *p* < 0.05) after adding rumination and reflection to the model (partial mediation). The specific indirect effect of the positive tendency to forgive on depression was significant through rumination (β = −0.071, CI_95%_ = [−0.110, −0.034]), but insignificant via reflection (see Appendix A).

Next, multiple mediation analyses showed that the total indirect effect of reduced unforgiveness on GHQ (and all subscales) through rumination and reflection was significant (β = −0.13, CI_95%_ = [−0.185, −0.088]) (Figure 2). Reduced unforgiveness was significantly negatively related to rumination (β = −0.42, *p* < 0.001) and insignificantly associated with reflection (β = 0.02; *p* = 0.73). Rumination was a positive, whereas reflection an insignificant, predictor of general health and its subscales. Reduced unforgiveness had a significant total effect on general health (β = −0.43, *p* < 0.001) and its subscales, and the direct effect was also significant after adding mediators, indicating partial mediation (for GHQ β = −0.29, *p* < 0.001). The specific indirect effect of reduced unforgiveness on GHQ trough rumination was significant (β = −0.13, CI_95%_ = [−0.182, −0.088]). In the case of somatic problems, such as anxiety/insomnia (β = −0.058, CI_95%_ = [−0.058, −0.034]), social dysfunction (β = −0.095, CI_95%_ = [− 0.146, −0.048], and depression (β = −0.134, CI_95%_ = [−0.181, −0.092]), the indirect effect through rumination was also significant (see Appendix A).

Next, multiple mediation analyses showed that the total indirect effects of reduced unforgiveness on GHQ (and all subscales) through rumination and reflection were significant in females. In males, the total indirect effects of reduced unforgiveness on general health, somatic symptoms, and anxiety were significant (see Table 2). The specific indirect effects of reduced unforgiveness on GHQ and all subscales through rumination were significant in women, but only for general health, and for somatic symptoms and anxiety in men. The specific indirect effects of reduced unforgiveness on GHQ and all subscales trough reflection were insignificant in females and males.

Next, multiple mediation analyses showed that the total indirect effects of positive forgiveness on GHQ (and all subscales) through rumination and reflection were significant in females. In males, the total indirect effect of positive forgiveness on somatic symptoms was significant (see Table 3). The specific indirect effects of positive forgiveness on GHQ and all subscales trough rumination were significant in females, but only for general health, somatic symptoms, and anxiety in males. The specific indirect effects of positive forgiveness on GHQ and all subscales trough reflection were insignificant in females and males.

Next, multiple mediation analyses showed that the total indirect effects of positive forgiveness on GHQ (and all subscales) through rumination and reflection were significant in women. In men, the total indirect effect of positive forgiveness on somatic symptoms was significant (see Table 4). The specific indirect effects of positive forgiveness on GHQ and all subscales trough rumination were significant in women, but only on general health, somatic symptoms, and anxiety in men. The specific indirect effects of positive forgiveness on GHQ and all subscales through reflection were insignificant in women and men.

### 3.2. Study 2

Forgiveness (total score) and both positive forgiveness and reduced unforgiveness showed positive relationships with general scores and the six subscales of psychological well-being; however, they were negatively linked to rumination. Additionally, reduced unforgiveness was negatively related to reflection. Rumination was inversely correlated with psychological well-being; the only positive correlation reported for reflection was with personal growth (see Table 5).

We tested multiple mediators simultaneously, with rumination and reflection as mediators between forgiveness (positive and reduced unforgiveness) and psychological well-being, with age and gender as covariates (see Figure 3 and Figure 4). Covariates were insignificant. All results shown are standardized. Multiple mediation analyses showed that the total indirect effect of positive forgiveness on psychological well-being (β = 0.14, CI_95%_ = [0.071, 0.238]), as well as on such aspects as autonomy (β = 0.02, CI_95%_ = [0.010, 0.047]), environmental mastery (β = 0.02, CI_95%_ = [0.006, 0.035]), positive relations with others (β = 0.03, CI_95%_ = [0.014, 0.048]), and self-acceptance (β = 0.05, CI_95%_ = [0.024, 0.075]), through rumination and reflection, was significant. As regards personal growth and purpose in life, the total indirect effect via rumination and reflection was insignificant. Positive forgiveness was significantly negatively related to rumination (β = −0.30, *p* < 0.001) and insignificantly associated with reflection (β = 0.09, *p* = 0.23). Positive forgiveness had a significant total effect on PWB (β = 0.46, *p* < 0.001) and all its six dimensions, and the direct effect (DE) was also significant after introducing mediators, indicating partial mediation, apart from autonomy, for which DE was insignificant (β = 0.09, *p* = 0.32), indicating full mediation via rumination.

Next, we checked whether rumination and reflection mediate the relationship between reduced unforgiveness and psychological well-being (see Figure 4) with its six dimensions (see Appendix A). Reduced unforgiveness was negatively associated with both mediators: rumination (β = −0.52, *p* < 0.001) and reflection (β = −0.19, *p* < 0.01). Reduced unforgiveness showed a significant indirect effect on psychological well-being (β = 0.13, CI_95%_ = [0.051, 0.226]) and self-acceptance (β = 0.02, CI_95%_ = [0.009, 0.026]) via rumination and reflection, and an indirect effect on autonomy (β = 0.01, CI_95%_ = [0.005 0.023]), environmental mastery (β = 0.03, CI_95%_ = [0.009, 0.053]), and positive relations with others (β = 0.04 CI_95%_ = [0.017, 0.061]), mediated mainly by rumination. Finally, reduced unforgiveness showed no indirect effect on personal growth and purpose in life.

The multiple mediation analyses showed that the total indirect effects of total forgiveness, positive forgiveness, and reduction of unforgiveness on psychological well-being through rumination and reflection were significant only in women. The specific indirect effects through rumination were significant only in women. The specific indirect effects through reflection were insignificant (see Table 6).

## 4. Discussion

The present study aimed to extend our previous research on the relationship between two dimensions of forgiveness, positive forgiveness and reduced unforgiveness, and health by exploring the mediating role of rumination and reflection. Across the two studies, we tested the mediation model, where rumination and reflection mediated the link between the tendency to forgive, both positive forgiveness and reduced unforgiveness, and health. In Study 1, we used physical health as the outcome variable. In Study 2, the outcome variable was mental health.

The results are largely consistent with the hypotheses. Our findings indicate that the general disposition to forgive and reduced unforgiveness are inversely related to unhealthy outcomes (somatic problems, anxiety, social dysfunction, and depression). On the other hand, forgiveness is positively related to mental health measured as well-being. This finding seems consistent with previous ones showing that forgiveness plays an important role in individuals’ mental health [4,6,12,15,16]. For example, Abu-Raiya and Ayten [51] indicated a positive relationship between motivation for revenge and avoidance and anxiety among Muslims. On the other hand, Kaleta and Mróz [2] showed the inverse link between reducing unforgiveness, positive forgiveness, and depression in psychotherapy outpatients.

Next, the data obtained here show rumination positively relates to unhealthy outcomes, consistent with previous findings by Yapan et al. [52]. Further, our analyses suggest rumination is a negative predictor of well-being, but reflection is only positively related to personal growth [40].

Our results demonstrate that rumination is a more effective mediator between forgiveness and physical and mental health. These findings are in line with the previous research indicating the mediating role of rumination [37,38,39]. Further, this outcome corresponds with the results of other studies showing that sleep quality is negatively related to rumination [53], and rumination is an important link between stress at work and disturbed sleep [40]. Additionally, individuals with high levels of self-rumination and a high information style had higher levels of depressive symptoms, according to Luyckx and colleagues [54].

The mediating effect of rumination established here aligns with previous studies indicating that rumination mediates the relationship between various variables and mental and physical health. Previous studies show that if an independent variable is negative, increased rumination leads to lower levels of physical and mental health [37]. Borawski [37] also pointed to the mediating role of rumination between loneliness and depressive symptoms, where loneliness and rumination lead to higher levels of depressive symptoms. However, if an independent variable is positive, then decreased levels of rumination result in higher levels of health and well-being [55]. For example, mindfulness training and meditation, by decreasing rumination levels, were found to lead to enhanced well-being [55]. This was also confirmed in a study by Beshart and Pourbohlool [56], where anger rumination was considered in two different relationships between the variables: when the independent variable was negative (anger) and positive (anger control), whereas the dependent variables were psychological well-being and psychological distress. The obtained results showed that anger is linked to lower levels of psychological well-being (higher levels of psychological distress) by increasing anger rumination. Conversely, anger control is related to higher levels of psychological well-being (lower levels of psychological distress) through decreasing anger rumination.

Our studies also show a similar role of rumination. We found that higher levels of positive forgiveness are linked to lower levels of health problems (GHQ) through decreasing rumination and increasing reflection. Higher levels of reduced unforgiveness are related to lower levels of health problems (GHQ) through decreasing rumination. On the other hand, higher levels of reduced unforgiveness are linked to higher levels of psychological well-being through decreasing rumination and increasing reflection. Further, higher levels of positive forgiveness are related to higher levels of psychological well-being through decreasing rumination. The results point to rumination as a more predictable and stable mediator when compared to reflection. What is more, in both studies, rumination played the role of a mediator for positive forgiveness, reduction of unforgiveness, and health. Reflection, on the other hand, played this role either when an independent variable was positive forgiveness (Study 1) or when the independent variable was reduced unforgiveness (Study 2). These results indicate the lower stability of reflection as a mediator.

The results obtained here are consistent with the Goal Progress Theory of rumination based on the Zeigarnik Effect [23]. According to the Zeigarnik Effect, a finished task needs not to be recalled. By forgiving, individuals close difficult situations they experience. Individuals with a higher disposition to forgive, who reduce negative thoughts, emotions, and motivations, and change their approach to the wrongdoer into a positive one, are more able to close the wrongdoing in their mind. Reduction of negative thoughts, emotions, or motivations and changing the approach to the wrongdoer into a positive one lead to the closure of the wrongdoing in one’s mind.

Ruminative thinking is not maintained. Therefore, forgiveness reduces rumination, which in turn translates into better physical and mental health. The obtained results align with the assumptions of the Response Styles Theory [24]. Rumination damages social relationships, supports hostility, and inhibits “letting go” and solving the problem. Forgiveness decreases the maladaptive influence of rumination and is conducive to higher well-being.

Understanding gender differences in the use of rumination and reflection to cope with wrongdoing and achieve well-being was deemed important. Women seem to have more often social–emotional roles and stress-related problems because of the role, and therefore, they are more overwhelmed by negative emotions and are dealing with rumination [24,27]. We found that rumination mediated the relationship between forgiveness and both mental and physical health, mostly in women, which suggests that men use different strategies so that influence health when they experience wrongdoing. Our outcomes are supported by previous studies showing that gender differences in the tendencies to use certain emotional regulation strategies in the respect of health [25,27]. Polanco-Roman et al. [26] found the association between stress-related symptoms and suicidal ideation was partially accounted for by rumination among females, but not among males. Additionally, reflection did not help to elucidate the association between stress-related symptoms and suicidal ideation for all genders health [25].

On the other hand, our analyzes showed reflection to be a non-stable mediator between forgiveness and health. This is consistent with previous studies pointing to the absence of any relationship between reflection and well-being [57], although Trapnell and Campbell [21] suggested a positive relationship between these variables.

Several limitations to the present study should be noted. First, it was a cross-sectional study, and as such, a causal pathway could not be established. Future studies should consider finding an alternative mechanism for explaining the relationship between the positive/negative dimension of forgiveness, health, and self-consciousness.

Second, this study was carried out in the Polish population; therefore, we do not know whether the trends identified here can be generalized to other populations. Future studies should include other populations, such as individuals from different cultural backgrounds and countries. Recent studies suggest there are differences between collectivistic and individualistic cultures in experiencing forgiveness, where higher levels of forgiveness are reported in collectivistic rather than individualistic cultures [58]. Another possibility would be to conduct research in specific groups, such as those with somatic illnesses or those experiencing specific offenses. For example, previous research indicates the development of depressive symptoms in adults who have experienced childhood maltreatment [59]. It could be interesting to determine the role of forgiveness and self-consciousness on the well-being and health in this group. Third, the tools used here measured only trends. Future studies should include methods assessing, e.g., episodic forgiveness. Previous studies indicate that episodic and dispositional forgiveness have different predictors and consequences [11,60]. Given a specific offense, contextual factors such as transgression severity, apology, and reparation will be controllable. This could show the intensity of ruminations due to the transgression severity and its importance for the forgiveness-healing link. Our results can be applied in constructing individual psychological therapy to enhance positive forgiveness by reflection or reduced unforgiveness by suppressing rumination, which will consequently lead to better mental health in society.

## 5. Conclusions

Our study added to prior research by showing the mediating role of rumination between forgiveness and physical and mental health. This study demonstrated that reduced unforgiveness and positive forgiveness lead to psychological well-being and lower levels of depressive symptomatology, by reducing rumination. This mechanism is revealed in females in particular.

## Figures and Tables

**Figure 1 ijerph-20-06229-f001:**
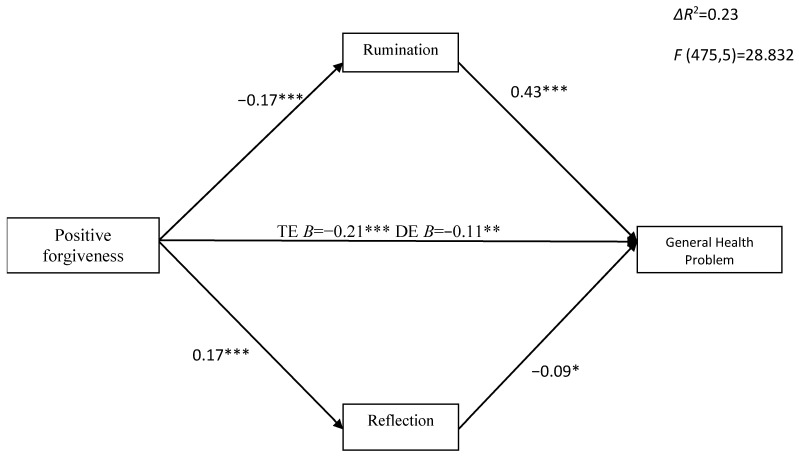
The indirect effect of positive forgiveness on general health problems via rumination and reflection. * *p* < 0.05; ** * p* < 0.01; *** *p* < 0.001; TE—total effect; DE—direct effect.

**Figure 2 ijerph-20-06229-f002:**
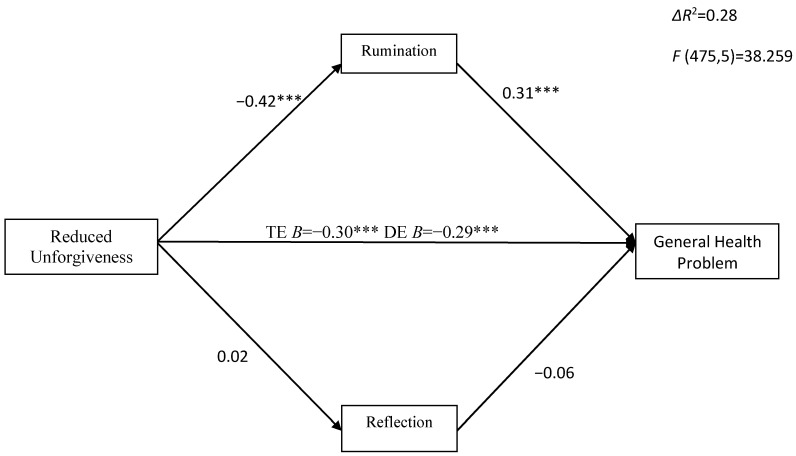
The indirect effect of reduced unforgiveness on general health problems via rumination and reflection. *** *p* < 0.001; TE—total effect; DE—direct effect.

**Figure 3 ijerph-20-06229-f003:**
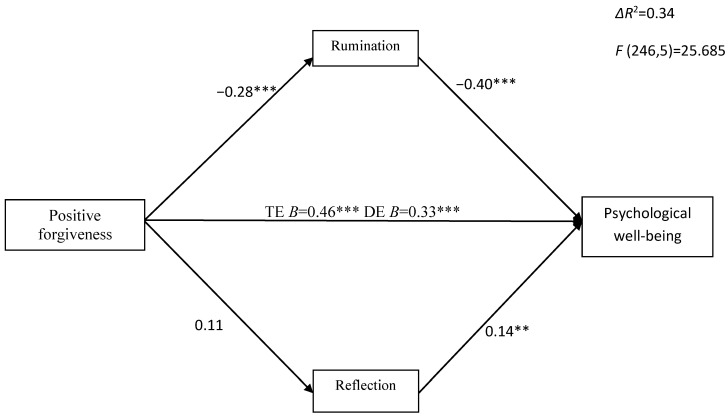
The indirect effect of positive forgiveness on psychological well-being via rumination and reflection. TE—total effect; DE—direct effect; ** *p* < 0.01; *** *p* < 0.001.

**Figure 4 ijerph-20-06229-f004:**
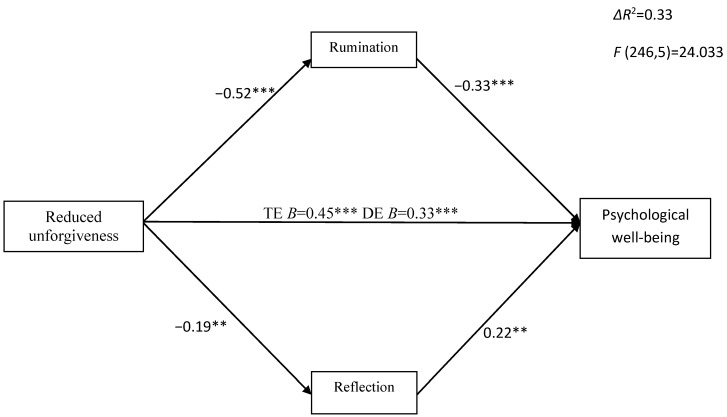
The indirect effect of reduced unforgiveness on psychological well-being via rumination and reflection. TE—total effect; DE—direct effect; ** *p* < 0.01; *** *p* < 0.001.

**Table 1 ijerph-20-06229-t001:** Means, standard deviations, and intercorrelations of Study 1 variables (*N* = 485).

		1	2	3	4	5	6	7	8	9	10
1	Forgiveness										
2	Positive Forgiveness	0.79 **									
3	Reduced unforgiveness	0.84 **	0.33 **								
4	Rumination	−0.42 **	−0.21 **	−0.45 **							
5	Reflection	0.09 *	0.15 **	0.00	0.33 **						
6	Somatic symptoms	−0.25 **	−0.12 **	−0.27 **	0.27 **	0.02					
7	Anxiety	−0.42 **	−0.22 **	−0.45 **	0.40 **	0.02	0.60 **				
8	Social dysfunction	−0.25 **	−0.12 **	−0.28 **	0.29 **	0.01	0.47 **	0.55 **			
9	Depression	−0.40 **	−0.21 **	−0.41 **	0.44 **	0.07	0.41 **	0.56 **	0.55 **		
10	GHQ_total	−0.42 **	−0.22 **	−0.45 **	0.44 **	0.04	0.77 **	0.86 **	0.77 **	0.79 **	
	*M*	80.14	41.86	38.28	17.55	21.49	14.68	15.31	14.84	10.24	55.01
	*SD*	13.96	7.96	9.11	5.03	5.61	4.13	4.77	3.37	4.36	13.41

* *p* < 0.05; ** *p* < 0.01.

**Table 2 ijerph-20-06229-t002:** The indirect effect of reduced unforgiveness on general health via rumination and reflection in females and males.

Dependent Variables	Gender	IE–Total		IE Rumination		IE Reflection	
		B	CI_95%_	B	CI_95%_	B	CI_95%_
GHQ_total	Females	−0.14	−0.199, −0.087	−0.14	−0.202, −0.089	0.00	−0.008, 0.013
	Males	−0.10	−0.204, −0.013	−0.10	−0.201 −0.027	0.00	−0.024, 0.032
Somatic symptoms	Females	−0.07	−0.135, −0.017	−0.07	−0.137, −0.017	0.00	−0.005, 0.008
	Males	−0.12	−0.223, −0.044	−0.12	−0.213, −0.048	0.00	−0.034, 0.019
Anxiety	Females	−0.11	−0.174, −0.063	−0.11	−0.176, −0.064	0.00	−0.009, 0.012
	Males	−0.09	−0.210, −0.004	−0.10	−0.204, −0.017	0.00	−0.023, 0.029
Social dysfunction	Females	−0.11	−0.170, −0.064	−0.11	−0.172, −0.066	0.00	−0.008, 0.014
	Males	0.00	−0.115, 0.116	0.00	−0.107, 0.100	0.00	−0.027, 0.035
Depression	Females	−0.15	−0.205, −0.101	−0.15	−0.208, −0.102	0.00	−0.007, 0.011
	Males	−0.06	−0.165, 0.020	0.00	−0.167, 0.003	0.01	−0.011, 0.037

**Table 3 ijerph-20-06229-t003:** The indirect effect of positive forgiveness on general health via rumination and reflection in females and males.

Dependent Variables	Gender	IE–Total		IE Rumination		IE Reflection	
		B	CI_95%_	B	CI_95%_	B	CI_95%_
GHQ total	Females	−0.10	−0.158, −0.054	−0.08	−0.137, −0.054	−0.01	−0.040, 0.000
	Males	−0.05	−0.131, 0.030	−0.04	−0.116, 0.031	−0.00	−0.054, 0.034
Somatic symptoms	Females	−0.05	−0.092, −0.010	−0.04	−0.082, −0.016	−0.00	−0.025, 0.014
	Males	−0.06	−0.157, −0.007	−0.04	−0.117, 0.027	−0.02	−0.083, 0.010
Anxiety	Females	−0.09	−0.145, −0.050	−0.07	−0.126, −0.037	−0.01	−0.041, 0.001
	Males	−0.05	−0.133, 0.030	−0.04	−0.116, 0.029	−0.01	−0.057, 0.360
Social dysfunction	Females	−0.08	−0.135, −0.044	−0.06	−0.111, −0.032	−0.01	−0.042, 0.001
	Males	−0.01	−0.082, 0.054	−0.01	−0.057, 0.017	−0.00	−0.057, 0.051
Depression	Females	−0.10	−0.157, −0.055	−0.09	−0.139, −0.042	−0.01	−0.036, 0.002
	Males	0.01	−0.082, 0.058	−0.03	−0.086, 0.022	0.01	−0.011, 0.061

**Table 4 ijerph-20-06229-t004:** The indirect effect of general forgiveness on general health via rumination and reflection in females and males.

Dependent Variables	Gender	IE–Total		IE Rumination		IE Reflection	
		B	CI_95%_	B	CI_95%_	B	CI_95%_
GHQ	Females	−0.14	−0.204, −0.087	−0.13	−0.194, −0.086	−0.00	−0.020, 0.003
	Males	−0.09	−0.190, −0.005	−0.09	−0.180, −0.030	0.00	−0.028, 0.043
Somatic symptoms	Females	−0.06	−0.130, −0.008	−0.06	−0.124, −0.012	0.00	−0.013, 0.009
	Males	−0.12	−0.222. −0.043	−0.11	−0.198, −0.044	−0.01	−0.061, 0.015
Anxiety	Females	−0.11	−0.175, −0.062	−0.11	−0.165, −0.062	0.00	−0.020, 0.004
	Males	−0.09	−0.187, −0.001	−0.09	−0.178, −0.026	0.00	−0.033, 0.044
Social dysfunction	Females	−0.12	−0.182, −0.068	−0.11	−0.172, −0.066	−0.01	−0.021, 0.004
	Males	−0.01	−0.118, 0.092	−0.01	−0.099, 0.059	0.00	−0.040, 0.053
Depression	Females	−0.15	−0.207, −0.098	−0.14	−0.199, −0.097	0.00	−0.019, 0.004
	Males	−0.03	−0.121, 0.047	−0.05	−0.127, 0.001	0.01	−0.005, 0.067

**Table 5 ijerph-20-06229-t005:** Means, standard deviations, and intercorrelations of study variables (*N* = 253).

		1	2	3	4	5	6	7	8	9	10	11	12
1	Forgiveness												
2	Positive Forgiveness	0.77 **											
3	Reduced unforgiveness	0.86 **	0.35 **										
4	Rumination	−0.52 **	−0.30 **	−0.53 **									
5	Reflection	−0.08	0.09	−0.20 **	0.25 **								
6	Psychological well-being	0.56 **	0.46 **	0.47 **	−0.46 **	0.07							
7	Autonomy	0.28 **	0.19 **	0.26 **	−0.29 **	0.06	0.63 **						
8	Environmental Mastery	0.36 **	0.35 **	0.25 **	−0.29 **	0.02	0.72 **	0.41 **					
9	Personal Growth	0.34 **	0.33 **	0.23 **	−0.14 *	0.21 **	0.59 **	0.24 **	0.34 **				
10	Positive Relations Others	0.47 **	0.38 **	0.39 **	−0.40 **	−0.01	0.78 **	0.36 **	0.44 **	0.34 **			
11	Purpose in Life	0.26 **	0.19 **	0.24 **	−0.19 **	0.07	0.57 **	0.18 **	0.21 **	0.38 **	0.40 **		
12	Self−Acceptance	0.57 **	0.44 **	0.50 **	−0.51 **	0.00	0.81 **	0.41 **	0.58 **	0.31 **	0.61 **	0.27 **	
	*M*	80.42	43.72	36.64	18.26	22.05	67.39	10.71	11.62	12.06	11.72	11.28	9.92
	*SD*	13.95	7.54	9.39	4.17	3.54	9.20	2.20	2.09	1.70	2.34	2.12	2.71

* *p* < 0.05; ** *p* < 0.01.

**Table 6 ijerph-20-06229-t006:** The indirect effect of forgiveness on psychological well-being via rumination and reflection in females and males.

Independent Variables	Gender	IE–Total		IE rumination		IE reflection	
		B	CI_95%_	B	CI_95%_	B	CI_95%_
Positive Forgiveness	Females	0.17	0.093; 0.263	0.14	0.070; 0.220	0.03	−0.003; 0.082
	Males	0.07	−0.060; 0.214	0.07	−0.034; 0.219	0.00	−0.056; 0.036
Reduced Unforgiveness	Females	0.15	0.071; 0.251	0.18	0.099; 0.277	−0.03	−0.075; 0.016
	Males	0.13	−0.042; 0.320	0.17	−0.011; 0.399	−0.04	−0.146; 0.067
Forgiveness	Females	0.16	0.086; 0.257	0.16	0.087; 0.254	0.00	−0.035; 0.046
	Males	0.08	−0.074; 0.288	0.11	−0.044; 0.335	−0.02	−0.110; 0.033

## Data Availability

The dataset presented in this study is available on reasonable request from the corresponding author.

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
