# Peer review of "Forgive, Let Go, and Stay Well! The Relationship between Forgiveness and Physical and Mental Health in Women and Men: The Mediating Role of Self-Consciousness"

_ijerph, 2023, doi:10.3390/ijerph20136229_

Round 1

Reviewer 1 Report

Dear authors

Thank you for submitting the paper "Forgive, Let Go, and Stay Well! Mediating Role of Self−Consciousness in the Relationship Between Forgiveness and Physical and Mental Health in Women and Men" for appreciation at the International Journal of Environmental Research and Public Health. The paper brings relevant information about how forgiveness can affect health and mental health.

Despite the fact that the topic is relevant for the health and mental health fields, the paper presents significant issues that might need to be revised, especially in the Materials and Methods section. Please see below:

  • The introduction does not explain the relationship between forgiveness and mental health, it mostly focuses on health

  • The manuscript does not present any information explaining the reasons for 2 studies.

  • The 2 studies had different inclusion and exclusion criteria. Apparently study 1 included the general population, while study 2 included individuals with serious adverse childhood experiences. For that reason, the results are not comparable.

  • The hypothesis presented do not mention mental health as an outcome, as it was stated in the discussion section

  • Insufficient details were given to explain the 2 studies, it is not clear which variables were tested in the two studies.

I suggest the authors could separate the studuies and present the results in 2 different manuscripts.

No significant issues with English language.

Author Response

Reviewer 1 

Thank you very much for your review. We appreciate your suggestions and comments, which will undoubtedly improve the quality of our manuscript.

The introduction does not explain the relationship between forgiveness and mental health, it mostly focuses on health

  • We have highlighted more the information concerning the areas of mental health (well-being, distress, depression).

The manuscript does not present any information explaining the reasons for 2 studies.

  • We added this information. “We verified the hypotheses in two studies, so that we could see whether self-consciousness would be a universal mediator, regardless of the type of health presented by the dependent variable.”

The 2 studies had different inclusion and exclusion criteria. Apparently study 1 included the general population, while study 2 included individuals with serious adverse childhood experiences. For that reason, the results are not comparable.

  • Thank you for this comment. Our aim was to verify whether self-consciousness is a universal mediator between forgiveness and healing or only under certain conditions. 

The hypothesis presented do not mention mental health as an outcome, as it was stated in the discussion section

  • We have completed the hypothesis.

Insufficient details were given to explain the 2 studies, it is not clear which variables were tested in the two studies.

  • We have added more specific information in section 2.2. Measures, and this information is included in description 2.1. separately for each study

I suggest the authors could separate the studies and present the results in 2 different manuscripts.

  • Thank you for your suggestion, however, we wanted to show the versatility of the mediation of self-consciousness, so we decided to combine these studies in one article

Reviewer 2 Report

This paper focuses on the relationship between forgiveness and health and it has some elements of innovation and originality. With a clear argumentation process and persuasive results, this paper successfully demonstrates self−consciousness as a mediator of the relationship between forgiveness and health. However, a few suggestions go as follows:

1. Title of the paper. The paper writes, “The purpose of this research was to test how positive forgiveness and reduced unforgiveness relate to health. Additionally, we explored selfconsciousness……”. However, the focus of this title is self-consciousness. It seems this title fails to reflect the dominant and sparking parts of this research.

2. The first hypothesis. The relationship between forgiveness and health has been repeatedly researched, roughly with consistent results worldwide; therefore, the justification for the necessity of research is nowhere to be found in this paper. Besides, it is recommended that a summary is added to highlight the research gap at the end of 1.2.

3. Participant selection. Since this study focuses on forgiveness, the adverse experience is essential to its success. Though it is described as an inclusion criterion, it is interesting to know how to ensure all participants had that adverse experience. How did you design participant screening in practice?

4. Grammatical errors. We considered it important to…… (line 478); such that people (line 54); We used in both study we used (line 12). It is suggested that the authors double check the grammar.

4. Grammatical errors. We considered it important to…… (line 478); such that people (line 54); We used in both study we used (line 12). It is suggested that the authors double check the grammar.

Author Response

Reviewer 2

Thank you very much for your review. We appreciate your suggestions and comments, which will undoubtedly improve the quality of our manuscript.

  1. 1. Title of the paper. The paper writes, “The purpose of this research was to test how positive forgiveness and reduced unforgiveness relate to health. Additionally, we explored self−consciousness……”. However, the focus of this title is self-consciousness. It seems this title fails to reflect the dominant and sparking parts of this research.

 Answer: We have changed the title so that it does not indicate the dominant role of the mediator:  Forgive, Let Go, and Stay Well! in The Relationship Between Forgiveness and Physical and Mental Health in Women and Men: The Mediating Role of Self−consciousness

  1. The first hypothesis. The relationship between forgiveness and health has been repeatedly researched, roughly with consistent results worldwide; therefore, the justification for the necessity of research is nowhere to be found in this paper. Besides, it is recommended that a summary is added to highlight the research gap at the end of 1.2.

 Answer: We felt it was important to explore the relationship between forgiveness and health before introducing the mediation hypothesis.  Therefore, we decided to introduce a hypothesis on the relationship between forgiveness and health.

We have added the missing information in the summary about the research gap.

  1. Participant selection. Since this study focuses on forgiveness, the adverse experience is essential to its success. Though it is described as an inclusion criterion, it is interesting to know how to ensure all participants had that adverse experience. How did you design participant screening in practice?

 Answer: We added information about design participant screening:   The  ACS was assessed via 11 questions with yes-no response categories designed for traumatic events during childhood. The participants were asked about neglect, physical/emotional abuse, and parental alcohol use. They were invited to continue the study if they answered yes to at least one question. 

  1. Grammatical errors. We considered it important to…… (line 478); such that people (line 54); We used in both study we used (line 12). It is suggested that the authors double check the grammar.

 Answer: We have corrected grammatical errors.

Reviewer 3 Report

The main research hypotheses are aptly identified in the introduction. Other literature is identified and how it relates to the current study. However, the following minor points need to be revised.

1. In the abstract, it is required to write more specifically about research methods such as the study subjects and data collection period. Also, it is required to describe that data collection was conducted through Study 1 and Study 2 in this study.

 2. If the variable name presented as 'general health' in Figures 1 and 2 is described with a negative word such as health problem, it will be easier for readers to understand the relationship between the variables.

Some minor language mistakes should be revised. 

Author Response

Reviewer 3 

Thank you very much for your review. We appreciate your suggestions and comments, which will undoubtedly improve the quality of our manuscript.

  1. In the abstract, it is required to write more specifically about research methods such as the study subjects and data collection period. Also, it is required to describe that data collection was conducted through Study 1 and Study 2 in this study.

Answer: We added some information in abstract.

  1. If the variable name presented as 'general health' in Figures 1 and 2 is described with a negative wordword, such as health problem, it will be easier for readers to understand the relationship between the variables.

Answer: We have changed the names in the figures.

Round 2

Reviewer 1 Report

Dear authors

I appreciate the clarifications and the revisions.

All the best,

The paper needs extensive English language review.

Reviewer 2 Report

 The authors have done a good revision. I recommended this paper to be published after a minor copyediting.

 The authors have done a good revision. I recommended this paper to be published after a minor copyediting.